# DNA-Binding Anticancer Drugs: One Target, Two Actions

**DOI:** 10.3390/molecules26030552

**Published:** 2021-01-21

**Authors:** Bruce C. Baguley, Catherine J. Drummond, Ying Yi Chen, Graeme J. Finlay

**Affiliations:** Auckland Cancer Society Research Centre, Faculty of Medical Sciences, The University of Auckland, Auckland 1023, New Zealand; cath.drummond@otago.ac.nz (C.J.D.); sawwiechan@gmail.com (Y.Y.C.); g.finlay@auckland.ac.nz (G.J.F.)

**Keywords:** DNA binding, antitumour, topoisomerase, cell cycle, pharmacokinetics, immunogenic cell death

## Abstract

Amsacrine, an anticancer drug first synthesised in 1970 by Professor Cain and colleagues, showed excellent preclinical activity and underwent clinical trial in 1978 under the auspices of the US National Cancer Institute, showing activity against acute lymphoblastic leukaemia. In 1984, the enzyme DNA topoisomerase II was identified as a molecular target for amsacrine, acting to poison this enzyme and to induce DNA double-strand breaks. One of the main challenges in the 1980s was to determine whether amsacrine analogues could be developed with activity against solid tumours. A multidisciplinary team was assembled in Auckland, and Professor Denny played a leading role in this approach. Among a large number of drugs developed in the programme, *N*-[2-(dimethylamino)-ethyl]-acridine-4-carboxamide (DACA), first synthesised by Professor Denny, showed excellent activity against a mouse lung adenocarcinoma. It underwent clinical trial, but dose escalation was prevented by ion channel toxicity. Subsequent work led to the DACA derivative SN 28049, which had increased potency and reduced ion channel toxicity. Mode of action studies suggested that both amsacrine and DACA target the enzyme DNA topoisomerase II but with a different balance of cellular consequences. As primarily a topoisomerase II poison, amsacrine acts to turn the enzyme into a DNA-damaging agent. As primarily topoisomerase II catalytic inhibitors, DACA and SN 28049 act to inhibit the segregation of daughter chromatids during anaphase. The balance between these two actions, one cell cycle phase specific and the other nonspecific, together with pharmacokinetic, cytokinetic and immunogenic considerations, provides links between the actions of acridine derivatives and anthracyclines such as doxorubicin. They also provide insights into the action of cytotoxic DNA-binding drugs.

## 1. Introduction

Cancer chemotherapy has gone through three phases over the last 70 years. The first was dominated by the development of cytotoxic drugs that damaged the genetic makeup of the cancer cell, the second was dominated by drug-specific signalling pathways and the third was associated with enhancing host immune responses to eliminate cancer cells. Despite its promise in precision medicine, targeted therapy is limited by the biology of cancer [1], and cytotoxic therapy has been recognised as important for the generation of host responses [2]. One particular focus of the latter has been the development of DNA-binding anticancer drugs. The elucidation of the structure of double-stranded DNA in 1953 by Watson, Crick and Wilkins [3] presented an early, structurally precise molecular target. It is instructive to retrace the logic behind the development of DNA-binding drugs and how they relate to precision cancer therapy. Studies at the Auckland Cancer Society Research Centre (ACSRC), initiated during the late 1960s under the directorship of Professor Bruce Cain, used molecular modelling to design drugs that could intercalate between adjacent DNA base pairs and disrupt the normal function of DNA, leading to anticancer activity. Synthesis of a series of 9-anilinoacridine derivatives identified compounds with significant activity against the L1210 transplantable murine leukaemia [4]. Professor Cain assembled a multidisciplinary team that eventually included medicinal chemists, molecular and cell biologists, pharmacologists and clinical oncologists. Amsacrine, one of a large number of 9-anilinoacridine derivatives synthesised and tested in this programme, was selected for testing in the US National Cancer Institute’s anticancer drug development programme. Amsacrine advanced to Phase I and Phase II clinical trials, was found to have significant activity against human acute leukaemia [5] and entered into worldwide use.

Over a similar time period, research on antibiotics had also led to the identification of DNA-binding drugs that demonstrated antitumour activity [6]. One of the most important was doxorubicin, an anthracycline antibiotic that was found to have utility against a variety of malignancies. A fascinating question to emerge from this early research was why amsacrine’s antitumour activity was limited to leukaemia while doxorubicin’s activity covered a broader spectrum. Subsequent research at the ACSRC has demonstrated the importance of embracing a number of scientific disciplines, including physics, chemistry, molecular biology, toxicology, pharmacology and immunology, in attempts to answer this question. This brief review commences with an outline of the molecular targets of amsacrine and how a study of amsacrine analogues led to the synthesis of *N*-[2-(dimethylamino)-ethyl]-acridine-4-carboxamide (DACA), which had novel features. Subsequent synthesis of new analogues with higher dose potency also helped to explain a toxic side-effect uncovered in the clinical trial of DACA. Other properties of DACA have enabled it to be linked to those of doxorubicin, emphasising the potential importance of host immunity in the actions of drugs that target topoisomerase II.

## 2. DNA Topoisomerases IIα and IIβ, the Molecular Targets of Amsacrine

One of the most important advances in our understanding of DNA function has been in the field of topology, where it has been shown that three-dimensional structures can be changed without alteration of the nucleotide sequence. DNA topology can be changed by enzymes known as topoisomerases, namely topoisomerase I, which changes the linking number (the number of times a linear polymer is twisted around itself) following reversible breakage of a single strand of the DNA double helix, and topoisomerases IIα and IIβ, which change the linking number following reversible breakage of both DNA strands. Topoisomerases IIα and IIβ have the striking property of being able to pass one double-stranded segment of DNA through another, an essential process in the maintenance of life. Early studies in several laboratories showed that amsacrine induced DNA damage in cultured cells [7,8], but a breakthrough came in 1984 with the finding that amsacrine acted as a poison for topoisomerase II enzymes [9] by inhibiting the religation of broken DNA strands. Doxorubicin was also found to have a similar action [10], thus linking the function of the two drug families, but these findings did not explain why doxorubicin was much more active against solid tumours than was amsacrine.

Professor Bill Denny joined the ACSRC in 1972 and made major contributions to the development of a large series of amsacrine analogues with the aim of determining the optimal structural features for antitumour activity. The development of assays for DNA-binding affinity and kinetics was complemented by the development of cell culture techniques, allowing multiple regression analyses to be carried out to analyse in vitro and in vivo antitumour activity in terms of lipophilic character, base strength and DNA-binding affinity [11]. The latter was found to be necessary but not sufficient for anticancer activity [12]. Molecular modelling suggested that while the acridine moiety was buried in the DNA double helix, the anilino moiety projected from the minor groove, raising the question of whether it might interact directly with the topoisomerase II enzyme.

## 3. The Discovery of *N*-[2-(Dimethylamino)-Ethyl]-Acridine-4-Carboxamide (DACA)

In the early 1980s, the ACSRC strategy for the in vivo testing of new DNA-binding drugs shifted from mouse leukaemias (L1210 and P388) to a mouse carcinoma [13], while still paying attention to DNA-binding affinity and physicochemical characteristics. Initially, the Lewis lung (3LL) tumour was used to screen for active drugs, and a large number of new analogues of amsacrine were synthesised and tested. One feature of these results was that some analogues containing a substituted carboxamide substituent on the acridine chromophore showed moderate activity against this lung carcinoma [14]. One derivative, asulacrine, was advanced to Phase I/II clinical trials but showed only modest activity against clinical carcinomas [15]. As synthesis of new analogues in the amsacrine series was being extended, Bruce Cain considered the question of whether the anilino side chain of amsacrine was absolutely required for antitumour activity, and he made the surprising finding that an amsacrine analogue that completely lacked an anilino side chain but contained an amino group on the 9-position and a dimethylaminoethylcarboxamide substituent on the 4-position of the acridine chromophore showed activity against the mouse leukaemia model [16].

Bill Denny took this concept further by considering the question of whether the 9-amino group that linked the anilino side chain to the acridine chromophore was necessary for activity, and he prepared a new compound, DACA, that lacked this group (Figure 1). DACA had only low activity against the P388 leukaemia but surprisingly, at the optimal dose, induced a 100% cure rate of mice carrying the Lewis lung tumour [17]. In structural terms, DACA resembled the anthracycline derivative doxorubicin in having a DNA-intercalating chromophore linked to a positively charged side chain, thus providing a potential link between the acridine and anthracycline series. DACA retained activity as a topoisomerase II poison, but this activity was weaker than that of amsacrine and was accompanied by inhibitory activity against topoisomerases I and II [18]. However, just as the anilino side chain of amsacrine was important for activity against the experimental leukaemia, the dimethylaminoethylcarboxamide side chain of DACA was important for activity against the experimental carcinoma; placement of this side chain at any other position of the acridine chromophore led to inactive compounds [17]. Important clues to the importance of this placement were provided by the results of studies by Wakelin, Denny and others, who showed that the dynamics of dissociation of DACA from the DNA-binding complex were strongly affected by the placement of the side chain [19,20].

## 4. Toxicological Studies with DACA

The activity of DACA against experimental murine solid tumours, together with evidence of activity in a human tumour xenograft [21], led to a Phase I trial of DACA under the auspices of what is now Cancer Research UK [22,23]. Studies during the dose-escalation phase uncovered an unexpected side effect: pain radiating from the intravenous infusion site. This was mild in some patients, but in others it was sufficiently intense to warrant immediate cessation of the infusion. Although unexpected, such behaviour had been recorded in some patients receiving intravenous doses of another DNA-binding drug, an anthrapyrazole derivative [23]. An important clue as to the cause of the toxicity was provided by the results of pharmacokinetic studies of DACA in mice, in which intraperitoneal doses were tolerated and had excellent antitumour activity but intravenous doses led to clonic seizures and death. The high degree of lipophilicity and the planar structure of DACA suggested interaction with lipids in ion channels, and its activity as an inhibitor of the human ether-à-go-go-related gene (hERG) was tested in patch clamp assays. At concentrations encountered in mice following intravenous administration, DACA showed evidence of hERG toxicity [24,25]. The focus of research was then altered to minimise this toxicity, i.e., to design drugs that were more dose potent than DACA as an antitumour agent but less potent than DACA as an ion channel inhibitor.

## 5. The Development of the DACA Analogue SN 28049

During the mid-1980s, changes were made in the ACSRC in vivo testing system because of animal ethical considerations; intravenous introduction of Lewis lung tumoursuspensions into mice led to the growth of tumours in the lung and to potential animal suffering. Subcutaneous implantation of tumour cell suspensions avoided this but led to ulcerating subcutaneous tumours, which were also ethically unacceptable. It was found that subcutaneous implantation of another murine tumour, the Colon 38 carcinoma (MCA38), was a more humane alternative, allowing growth delays of subcutaneous tumours to be measured using callipers. DACA was active but not curative against this tumour, meaning that it was a good system in which to search for more active analogues. Initially, DACA analogues in which the acridine chromophore was replaced by phenazine, phenylquinoline or other groups were tested [26], but none were more active against Colon 38 tumours than DACA alone. During this time, Bill Denny had initiated a collaboration with Professor Les Deady at Latrobe University in Melbourne to develop new analogues of DACA; chemical synthesis was carried out in Melbourne while the biological evaluation was carried out at the ACSRC. Analogues containing novel chromophores were developed and tested for DNA affinity, in vitro inhibition of tumour cell proliferation and in vivo growth delay of Colon 38 tumours. Most derivatives had lower in vivo activity than DACA, but some showed unexpectedly high activity [27]. One compound in particular, called SN 28049, showed both higher DNA-binding affinity and higher in vivo dose potency than DACA, and importantly showed lower hERG toxicity than DACA [24]. Significantly, SN 28049, along with some of its derivatives, was found to have curative activity against the Colon 38 tumour [27].

## 6. Insights from In Vitro Studies Using Colon Carcinoma Lines

DACA and SN 28049 marked important milestones in the drug development programme but again raised questions about the basis for their activity against solid tumour models. In order to gain further insights into this question, studies were carried out using cultured tumour cell lines. The HCT116 human colon carcinoma line, generously provided by Dr. B. Vogelstein, was chosen for study because it was available in two forms, one of which lacked the p53 pathway; this enabled assessment of the role of the p53 pathway in cellular responses to this drug [28]. An initial study revealed an unexpected result: short-term (one hour) exposure of cells to SN 28049 had a minimal effect on survival, as measured by clonogenic assay, but as the period of drug exposure increased, so did the cytotoxic activity as measured by clonogenicity [29]. The cellular response to this drug was thus more like that of cell-cycle-selective drugs, for which cytotoxicity was manifested only at a particular phase of the cell division cycle [29]. This presented an apparent paradox because topoisomerase II activity is generally maintained throughout the cell cycle. Flow cytometry studies provided a further unexpected and puzzling result: although SN 28049 induced the p53 pathway and inhibited cell proliferation in a p53-dependent manner, it was unable to induce DNA damage in G1-phase cells and to block them from entering S phase. It was unable to block mitosis but could prevent progress from mitosis to cell division in a p53-dependent way. Further work showed that it affected cells when they were close to the anaphase stage of the cell division cycle, inducing the appearance of a high proportion of binucleate cells and a smaller proportion of multinucleate cells [29].

Since the Colon 38 tumour helped to define the in vivo activity of SN 28049, it was of interest to investigate its in vitro response and to compare it with that of the HCT116 line [25]. A cell line derived from the Colon 38 tumour was therefore developed and its response to SN 28049 was studied using flow cytometry and other techniques. Like HCT116, the Colon 38 line responded in a p53-dependent manner by arresting late in the cell division cycle, showing an accumulation of binucleate and multinucleate cells. However, unlike HCT116, the Colon 38 line also showed evidence of p53- and p21-mediated cell cycle arrest in G1 phase.

## 7. Role of Topoisomerase II

A possible reason for the above discrepancy between the responses of the two cell lines is that topoisomerase II behaves differently in the HCT116 line. In proliferating cells, topoisomerase is known to have two functions that are essential for survival: the removal of DNA supercoils during DNA replication in S phase and the segregation of daughter chromatids during anaphase (Figure 2). DNA-binding drugs can compromise the first process by so-called poisoning, in which inhibition of the religation step essentially converts the DNA-linked topoisomerase into a DNA-damaging agent, leading to the induction of DNA double-strand breaks. They can also compromise the second process by inhibiting the catalytic function of topoisomerase during anaphase. The HCT116 cell line was developed from a tumour that, although not exhibiting chromatin instability, contains a frameshift mutation at exon 17 of the topoisomerase II alpha gene, corresponding to a region of the enzyme concerned with DNA breakage and reunion [30]. It is therefore possible that this mutation attenuates the ability of topoisomerase II to generate DNA double-strand breaks in response to SN 28049 but preserves its ability to facilitate topoisomerase II-dependent chromatid segregation during anaphase. The two processes differ in their dependence on topoisomerase II activity; in the first, increased activity leads to greater topoisomerase-induced DNA breakage activity and a larger cytotoxic effect, while in the second, decreased activity may, like inhibitors of the catalytic function of topoisomerase II, induce abnormalities at anaphase and compromise accurate segregation of chromatids [31,32,33,34].

Differing relative activity on each of these actions on topoisomerase II might therefore have more general implications, suggesting two broad classes of topoisomerase interaction with DNA-binding drugs. In the case of amsacrine, cytotoxicity may occur mainly as a consequence of topoisomerase poisoning, while in the case of SN 28049, DACA and doxorubicin, cytotoxicity may also occur as a consequence of faulty chromatid segregation. Both actions can result in activation of the p53 pathway but through different mechanisms; the first by activating the DNA damage response [35] and the second by acting on histone targets [36]. The properties of the HCT116 line potentially allow it to be used in the further study of the events around dissection of the two functions of the enzyme. One feature of this second action is that it may also involve dual inhibition of topoisomerases I and II. This was found with SN 28049, DACA and the anthracycline derivative aclarubicin [37]. Such activity could be important for chromatid segregation, linking the actions of the DACA series to those of the anthracyclines.

## 8. Pharmacokinetic Considerations in the Action of SN 28049

One of the unexpected findings regarding the in vivo response of Colon 38 tumours was that while SN 28049 had curative activity, etoposide, a topoisomerase II poison inducing comparable DNA breakage, was almost inactive [32]. In comparing the in vivo action of these two drugs, it was found that the drug-induced changes in the histological appearance of tumours exposed to SN 28049 persisted for many hours, while changes in tumours exposed to etoposide reversed rapidly. A potential explanation for this behaviour was that SN 28049 had a long tumour tissue half-life; several pharmacokinetic studies were carried out to investigate this possibility. While plasma and normal tissue SN 28049 concentrations decreased with a half-life of approximately 3 h, tumour tissue concentrations declined much more slowly [38]. Investigation of the cause of this behaviour using the Colon 38-derived cell line showed that the drug was sequestered in cytoplasmic vesicles in some kind of “depot” form. Such behaviour was similar to that reported for the drug doxorubicin [39], suggesting that cellular and tissue pharmacokinetics might help to explain both the experimental antitumour activity of SN 28049 and the clinical activity of doxorubicin.

One of the key factors in the activity of any anticancer drug is its ability to be delivered for the right duration and at the right concentration to tumour tissue. Of particular relevance is the consideration of whether the drug is active at all stages of the cell division cycle. In the case of amsacrine, its ability to generate double-stranded DNA breaks is related to topoisomerase II activity, which is generally present at all stages of the cell cycle but increases in S and G2 phases [40]. In the case of SN 28049, antitumour activity will be related partially to its effect as a topoisomerase II poison and partially to its ability to generate DNA damage through chromatid segregation defects at anaphase. The latter will be highly cell cycle phase dependent, and its exploitation requires drug concentrations to be maintained for a period of at least one cell cycle time.

## 9. Immunological Considerations in the Action of SN 28049

The proposal that doxorubicin may exert part of its antitumour action by direct cytotoxicity and partly by stimulating host innate immunity has attracted substantial interest [41]. The underlying principle is that DNA damage induced by cytotoxic drugs such as doxorubicin induces a cellular response that is related to an antiviral response. One of the best characterised of these responses is the STING response, in which the presence of cytoplasmic DNA leads to the induction of type I interferons and eventually to the development of a T-cell-mediated immune response, leading to tumour cell death [41]. Dying tumour cells may undergo so-called “immunogenic” cell death, whereby they release a variety of signals, such as the protein HMGB1, calcium ion and calreticulin, which act as markers of this process. Such a response has been well documented in the case of doxorubicin [41], and our preliminary experiments have shown that treatment of mice with SN 29049 also leads to increases in these markers. Cytoplasmic DNA arising as a consequence of a failure of chromatid segregation, arising in turn from the action of drugs such as SN 28049 and doxorubicin, could be an important activator of STING and perhaps of other immune pathways.

## 10. Perspective

A combination of chemical synthesis and biological evaluation, conducted primarily in the ACSRC and championed by Bill Denny and colleagues, has led to the elaboration of a large series of DNA-binding antitumour drugs. In this approach, the synthesis of a large series of related compounds with a range of physicochemical properties could be complemented by the application of biological assays to explore structure–activity relationships, thus optimising antitumour activity. The design of an effective clinical antitumour drug entails modification of physicochemical properties such as lipophilicity and base strength, important for protein binding and metabolism, as well as optimisation of DNA-binding affinity and binding kinetics. Minimisation of ion-channel-mediated toxicity is an important consideration, and maximisation of induced host immunity may ultimately be the most important feature of a successful candidate drug; current attention on cancer treatment is focussed on the suppression of immune checkpoints.

DNA interacts with many different cellular proteins, but its interaction with topoisomerase II is particularly important for cell survival. As indicated in Figure 2, two major topoisomerase II-dependent processes are essential for cell proliferation, namely the accurate replication of the genome and the accurate segregation of daughter chromatids prior to cell division. DNA-binding drugs can compromise the first process by so-called poisoning, in which inhibition of DNA religation leads to the induction of DNA double-strand breaks. They can also compromise the second process by inhibiting the catalytic function of topoisomerases IIα and IIβ during anaphase. The two processes differ in cell cycle selectivity; the first occurs throughout interphase while the second occurs selectively in anaphase. The picture that has built up around the results of the ACSRC programme is that, while all of the DNA-binding drugs in the series may affect both processes, the balance between the effects on the two processes can be altered by drug design. Moreover, the response of individual tumours may depend on the activity of topoisomerases IIα and IIβ; high activity may favour poisoning while low activity may favour damage in anaphase.

In conclusion, the efficacy of this class of DNA-binding drugs, like that of anthracycline-based anticancer drugs, is likely to involve tumour–host interactions as well as intrinsic cellular processes and targeted therapy. It will be necessary to carry out further research on pathways such as STING, which links damaged DNA to activation of the immune system. A promising avenue for future research involves the development of improved assays to assess the ability of this series of drugs to modulate immune-signalling networks.

## Figures and Tables

**Figure 1 molecules-26-00552-f001:**
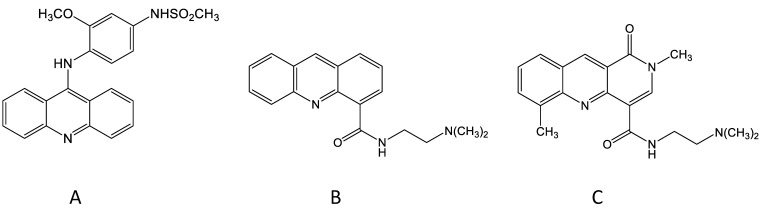
Chemical structures of amsacrine (**A**), *N*-[2-(dimethylamino)-ethyl]-acridine-4-carboxamide (DACA) (**B**) and SN 28049 (**C**).

**Figure 2 molecules-26-00552-f002:**
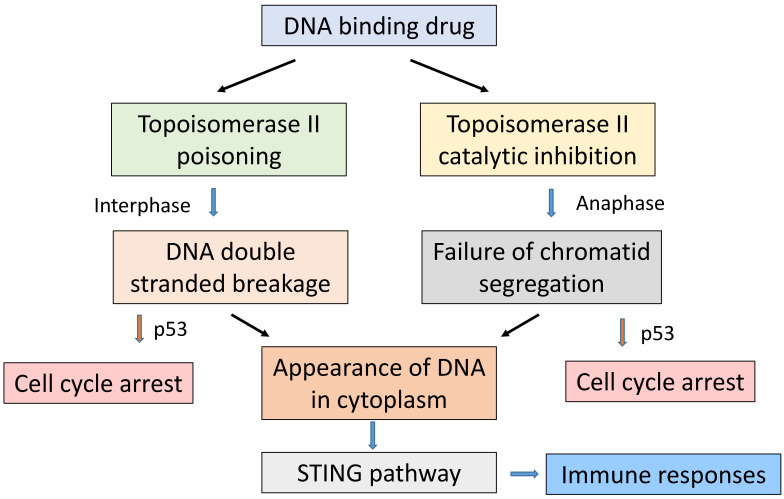
A simplified scheme that illustrates some of the main points of this review. Two main pathways link DNA binding, topoisomerase II action and drug-induced tumour cell death. In the first, which occurs broadly through most phases of the cell division cycle, inhibition of the religation step during strand passing leads to DNA double-strand breaks. In the second pathway, inhibition of DNA strand passing during anaphase interferes with the proper segregation of chromatids following mitosis. Both processes can activate the p53 pathway, leading to cycle arrest mainly in G1 phase for the first process and tetraploid G1 phase for the second. Both processes can lead to cell death and the activation of immune responses, including the stimulation of interferon genes (STING), induction of cytokines, induction of T-cell-mediated responses and cytotoxicity.

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
