# Peer review of "DNA-Binding Anticancer Drugs: One Target, Two Actions"

_molecules, 2021, doi:10.3390/molecules26030552_

Round 1

Reviewer 1 Report

A wonderful review of the history and logic behind the development of DNA-acting drugs, and the role of new new techniques to aid understanding of their biological responses.

  1. The manuscript provides a detailed history of the development of a new class of DNA-acting anticancer agents.  It included a summary of many structure-activity relationships that guided the drug development strategy.  It also provided a clear case for the dual role of the drug (one cycle-specific and the other not) even though the drug had the same DNA target for both mechanisms of action.
  2. The strength of the article is the clarity and extensive documentation of the presentation, coupled with an excellent figure to summarise the two actions of the drugs. The article provides an excellent explanation for how a drug can have two apparently contradictory mechanisms of action, one cell cycle specific and the other independent of the cell cycle.

I have no suggestions for any improvement to this high quality article.

Author Response

Thank you.

Reviewer 2 Report

The authors gave a very interesting topic on DNA binding anticancer drugs that have two actions. The two actions will improve the therapeutic effect significantly and will prompt the drug discovery of potent anticancer candidates. The manuscript should be accepted surely after addressing the following issues.

  1. As a review paper, there are only 42 references that are quite inadequate. In addition, the references are relatively old and cannot reflect the recent progress on this topic. I recommend a few related publications to be cited.

(I). “Design, synthesis and biological evaluation of novel phthalazinone acridine derivatives as dual PARP and Topo inhibitors for potential anticancer agents”, Chinese Chemical Letters, 2020, 31(2), 404-408.

(II). “Evodiamine-inspired dual inhibitors of histone deacetylase 1 (HDAC1) and topoisomerase 2 (TOP2) with potent antitumor activity”, Acta Pharmaceutica Sinica B, 2020, 10(7), 1294-1308.

  1. In the abstract, the sentence “A programme in the Auckland Cancer Society Research Centre, initiated by the late Professor Bruce Cain in the late 1960’s, aimed to develop DNA-binding drugs with clinical anticancer activity” seems non-relevant to this manuscript.

3. At the end of the introduction, the authors are suggested to give a summarized sentence to introduce the skeleton of the review.

Author Response

We thank the reviewer for these helpful comments.

We have added several more recent references, particularly in the area of the catalytic inhibition of topoisomerase action, which is the most novel area of the review and should allow the reader to access the literature more effectively. It should be noted that since much of the review involves references to the work of Professor Denny, these are comparatively old. We have not included the two references suggested by the reviewer because they lead to areas not covered in the review and would demand a wide number of further changes.

We have removed the first sentence in the abstract and have modified the second sentence.

We have added a section at the end of the introduction to provide a skeleton of the review.